# Identification and Characterization of Multiple Myeloma Stem Cell-Like Cells

**DOI:** 10.3390/cancers13143523

**Published:** 2021-07-14

**Authors:** Wancheng Guo, Haiqin Wang, Peng Chen, Xiaokai Shen, Boxin Zhang, Jing Liu, Hongling Peng, Xiaojuan Xiao

**Affiliations:** 1Department of Hematology, the Second Xiangya Hospital, Molecular Biology Research Center, School of Life Sciences, Hunan Province Key Laboratory of Basic and Applied Hematology, Central South University, Changsha 410011, China; 2204160215@csu.edu.cn (W.G.); 202501023@csu.edu.cn (H.W.); liujing2018@csu.edu.cn (J.L.); 2Xiangya Medical School, Central South University, Changsha 410013, China; 2204160219@csu.edu.cn (P.C.); 2204160222@csu.edu.cn (X.S.); 2204160218@csu.edu.cn (B.Z.)

**Keywords:** multiple myeloma, MM stem cell-like cells, marker, pathway

## Abstract

**Simple Summary:**

Although the existence of multiple myeloma (MM) cancer stem cells is controversial, a large number of studies have shown that there is a kind of cells in multiple myeloma that has stronger proliferation, migration, tumorigenesis, and drug resistance than general tumor cells, and so we call them the multiple myeloma stem cell-like population. Therefore, the definition, screening, and targeted inhibition of these cells are very important. In this paper, we list the markers used to screen the MM stem cell-like population and the research of related inhibitors in MM treatment and introduce the pathways related to stemness in MM and the main target molecules in these pathways.

**Abstract:**

Multiple myeloma (MM) is a B-cell tumor of the blood system with high incidence and poor prognosis. With a further understanding of the pathogenesis of MM and the bone marrow microenvironment, a variety of adjuvant cell therapies and new drugs have been developed. However, the drug resistance and high relapse rate of MM have not been fundamentally resolved. Studies have shown that, in patients with MM, there is a type of poorly differentiated progenitor cell (MM stem cell-like cells, MMSCs). Although there is no recognized standard for identification and classification, it is confirmed that they are closely related to the drug resistance and relapse of MM. This article therefore systematically summarizes the latest developments in MMSCs with possible markers of MMSCs, introduces the mechanism of how MMSCs work in MM resistance and recurrence, and discusses the active pathways that related to stemness of MM.

## 1. Introduction

Existing statistics show that the incidence of MM is the second-highest of the blood system tumors. Nowadays, the median age at diagnosis is 70 years; 37% of patients are younger than 65 years, and very few patients are younger than 30 years old [1,2]. However, MM is still an incurable disease. As the proportion of the elderly population rises, the morbidity and mortality of MM gradually increase [3,4]. Besides, the drug resistance and recurrence of MM are becoming increasingly prominent in clinical practice, which is closely related to the stem cell-like population in multiple myeloma cells.

About four decades ago [5], it was supposed that there was a cell population closely related to cancer genesis and self-renewal, and so they were called cancer stem cells (CSCs). More and more characteristics of CSCs have been discovered recently. The current study shows that CSCs are a kind of cell that determines the occurrence, development, and metastasis of tumor tissue and has three characteristics: self-renewal, differentiation, and infinite proliferation.

Although the existence and identification of MM stem cells are controversial at present, the existing data show that there is a kind of progenitor cell with low differentiation in the MM cell group that has the characteristics of CSCs and so we call them ‘MM stem cell-like cells’. The current identification of such cells depends on the detection of several surface markers. Due to the multi-directional differentiation and self-renewal characteristics of this cell, there is evidence that they are closely related to the relapse and drug resistance of MM [6,7]. Therefore, further research on the characteristics of MMSCs is likely to provide a theoretical basis for solving its high recurrence and drug resistance. The following article will introduce the potential surface molecular markers to identify MMSCs, the mechanism of drug resistance and recurrence of MM and therapies targeting CSC, and the activation of the common pathways related to the stemness of MM cells.

## 2. Markers of MMSCs and Drugs Targeting Them

### 2.1. Side Population Cell

Side population cells (SP cells) were first discovered by Goodell et al. They can excrete Hoechst 33342, so they show low staining and are distributed on one side of the main population cells [8]. SP cells’ excreting dyes are related to ABC transporters (especially ABCG2) [9], which are a series of membrane transport proteins that hydrolyze ATP transmembrane transport substrates and are expressed in a variety of CSCs, resulting in drug resistance in cancers such as esophageal cancer and oral squamous cell carcinoma [10,11]. According to the CSC theory, a small number of cancer stem cells hidden in cancer also promote tumor growth, which explains some clinical phenomena, such as the almost inevitable recurrence of tumors after the initially successful chemotherapy and/or radiotherapy, the phenomenon of dormancy and the metastasis of tumors [5]. Among MM cells, SP cells in G0/G1 phases are significantly more numerous than NSP cells [12,13], which indicates that the proportion of cells in the resting phase is higher in SP cells. SP cells from myeloma cell lines (KMS11, OPM1, RPMI-8226, U266, et al.) have high clonogenicity, tumorigenicity, and self-renewal ability [14,15]. The induced pluripotent stem cell (iPS)/embryonic stem cells (ES) genes (*SOX2, OCT4, NANOG, KLF4*, et al.) are highly expressed in SP cells [16,17]. In SP cells, drug efflux pump genes *ABCG2* and *ABCC3* are highly expressed in MM cells [14,18]. In short, the above results show that SP cells are closely related to the stemness, drug resistance, and recurrence of MM, so SP cells are often used as materials to replace MM stem cells.

Moreover, the existence of SP cells is closely related to the microenvironment of MM cells. Under certain microenvironmental conditions, SP cells will undergo significant changes. Myeloma BMSCs can generate a microenvironment that supports myeloma stem cells through the CXCR4 signaling pathway, the result of which is that, when growing on myeloma BMSCs, the percentage of SP cells is higher than that of control BMSCs, and SP cells have stronger clone formation ability than control BMSCs [16,19]. Granulocytic- myeloid-derived suppressor cells (G-MDSCs) can enhance the side population, sphere formation, and the expression of core genes of CSCs in MM cells through promoting DNA methylation [20]. Interestingly, the percentage of SP cells is related to culture conditions, such as culture time and hypoxic environment [16,21]. Surprisingly, the NSP cell population can be transformed into SP cells under a specific environment (such as hypoxia), which is regulated by the TGF-β1 pathway [14]. It is speculated that SP cells cannot completely represent the stem cells of MM, and there may be a small number of stem cells among NSP cells.

SP cells are also used to study the target of MM stem cell-like population or to verify MM stem cell markers. In Minjie Gao et al., Affymetrix microarrays were performed on 7-paired light-chain (LC) restricted SP (LC/SP) and bulk MM cells (CD138^+^) on 14 primary MM samples. After microarray analysis, CD24 was highly expressed in SP cells, and subsequent experiments proved that it is an important marker of MM stem cells [22]. The microarray was combined with other MM databases for biometric analysis and provided reliable molecular biological markers for the screening, prognosis, and new therapeutic targets of myeloma LC/SP cells [23,24]. Besides, from a high throughput assay, 19 Cancer testis antigen (CTA) genes were upregulated in the SP of MM, suggesting that CTA may be a target for MM stem cell-specific immunotherapy [25].

Drugs targeting SP cells have been developed a lot over the years. In the flow cytometric detection of SP cells, verapamil and reserpine were used as negative controls to inhibit the SP cells phenotype [16,26]. Proteasome inhibitors bortezomib (BTZ), carfilzomib, and ixazomib significantly suppressed the proliferation of SP cells in MM patient cells and MM cell lines, and metformin enhances the anti-SP effect of BTZ. Lenalidomide targets clonogenic SP cells in MM, rather than thalidomide [14]. EZH1/2 double inhibitor (or-s1) can effectively eradicate SP cells [27]. GSK126 can target EZH2 (epigenetic regulator), and it can kill the myeloma stem cell-like population by ALDH and SP analysis [28]. Fenretinide has a scavenging effect on MM-SP and NSP cells [29]. SP cells can also be reduced by c3b3 (a new diabody) that can inhibit pluripotency-related transcription factors (*SOX2*, *OCT4*, *NANOG*, etc.) [12]. It is found that S1-401 (drug targeting IL-3) can improve the prognosis of MM patients, and the growth of SP cells is inhibited [30]. Hucd26mab can reduce the proportion of SP cells in CD26^+^ MM cells [31]. The combination of anti-ABCG2 monoclonal antibody EPI MBS+mAb and ultrasound treatment can reduce the clonal ability of MM CD138^−^ CD34^−^ CSC isolated from a human MM RPMI 8226 cell line and inhibit tumorigenesis in nude mice [32]. In addition, natural products are considered biocompatible and reliable treatments for human cancer [33,34]. Diallyl thiosulfinatecan enhances the inhibitory effect of dexamethasone on SP cells [35]. Baicalein can downregulate ABCG2 expression and suppress SP cells [36,37]. Many types of drugs can significantly inhibit the growth of SP cells in MM, which provides a research basis for strategies targeting MMSCs.

### 2.2. ALDH

ALDH is an intracellular enzyme that participates in the detoxification and differentiation of cells through oxidation and plays a role in the development of drug resistance in cancer cells. In CSCs, ALDH can regulate the differentiation, apoptosis, and growth of CSC and participate in drug resistance [38]. In addition, ALDH is highly active in leukemia and solid tumors of the colon, breast, brain, prostate, pancreas, and ovary and is likely to be a target for cancer diagnosis and treatment [26,39]. These characteristics of ALDH show that it has certain significance in the identification of MM stem cells.

ALDH1^+^ MM cells contribute to MM stemness and resistance. ALDH1^+^ MM cells have higher colony-forming ability than ALDH1^−^ cells, and their tumorigenic rate is significantly higher in NOD/SCID mice, indicating that ALDH1^+^ cells have tumorigenic stem cell-like features [40]. ALDH1A1 is the dominant isoform in MM; when the expression of ALDH1A1 in ARP1 and OPM1 cells was driven by a lentivirus vector, the cells showed resistance to bortezomib and adriamycin [41]. In BTZ-resistant (ANBL6-BR) cells, ALDH1^+^ cells constitute a larger proportion of the population than in wild-type (ANBL6) cells, and ALDH1A1 is highly expressed in BTZ-resistant cells [42]. ALDH-activity detection has been used to reflect the stemness of MM cells in the study of drug killing-effect detection, surface markers, and characteristic proteins identification in the MM stem cell-like population [18,43,44,45]

Targeting ALDH^+^ cells is a new strategy for the treatment of acute myeloid leukemia (AML) and breast cancer [46,47]. For example, in the CD34^+^ CD38^−^ phenotype of AML cells and pre-leukemic stem cells, ALDH is often highly expressed, which can be inhibited by the new isatin analog KS99 [48]. It has been reported that ALDH inhibitor disulfiram acts in a copper-dependent manner on ALDH1A1 and Hedgehog transcription factors GLI1 and GLI2 to clear ALDH-positive MM cells [49]. Interestingly, 5-fluoro-2′- deoxyuridine (FdUrd) stimulation led to decreased activity of ALDH. In addition, lycorine can suppress the Wnt/β-catenin pathway and has a cytotoxicity effect on MM ALDH1^+^ cells. Additionally, wee1 kinase inhibitor MK1775 can significantly decrease the rate of ALDH1^+^ cells. Surprisingly, proteasome inhibitor BTZ can enhance the percent of ALDH1^+^ cells in glioblastoma, synovial sarcoma, pancreatic adenocarcinoma, and MM cells [42,50]. Furthermore, lycorine or wee1 kinase inhibitor MK1775 can decrease the rate of ALDH1^+^ cells in combination with BTZ [35,42]. However, the role of immunomodulators, other proteasome inhibitors, and other MM clinical drugs in ALDH has not yet been reported. This is something that future research can focus on.

### 2.3. CD138

CD138 (syndecan-1) is a kind of cell-adhesion molecule, whose deletion can lead to the loss of contact inhibition in some cells. CD138 was absent in poorly differentiated B cells and highly expressed in terminally differentiated B cells [51,52,53]. In vivo and in vitro experiments showed that MM cells derived from CD138^−^ cells had stronger clone-forming ability and stronger stemness characteristics than CD138^+^ cells, which was related to the drug resistance of MM [54,55]. As early as 2003, William et al. found that whether in MM cell lines (RPMI 8226 and NCI-H929) or clinical samples of MM patients, MM cells with CD138^−^ had stronger colony-forming and proliferation ability [56]. In the CD138^−^ MM group, the downregulation of CD229 could reduce the ability of MM cell colony formation and enhance the effect of chemotherapy [57]. CD138^−^ALDH1^+^ MM cells have strong colony-forming and tumor initiation ability [58]. In addition, some scholars further explored the effect of SH3GL3 on the migration and invasion of the CD138^−^ MM stem cell-like population [59]. Recently, Phoebe et al. described the role of interferon regulatory factor 4 (IRF4) in the regeneration of myeloma progenitor cells in cell stem cell and found that the overexpression of IRF4 increased the proportion of myeloma progenitor cells, among which CD138^−^ is used as an important screening marker for myeloma progenitor cells [60].

### 2.4. CD24

CD24 antigen is a cell-adhesion protein linked to glycosylphosphatidylinositol (GPI), which can mediate the antigen-dependent activation and proliferation of B cells. CD24 has high expression in many types of tumors and promotes cancer invasion and metastasis. CD24^+^ has already been regarded as a CSC marker in ovarian cancer [61] and liver cancer [62], while CD24^−^ has been found in breast CSCs [63]. A recent study showed that the tumor-initiating cells in multiple myeloma can be identified by CD24 [22]. In SP cells or MM cells in patients after treatment, CD24 has higher expression level in the SP cells of MM patients and MM patients after treatment, and high expression of CD24 indicates poor prognosis in MM patients. In addition, CD24^−^-positive cells have stronger clone formation ability and tumorigenicity, higher iPS/ES gene expression, and stronger resistance to MM clinical drugs BTZ, carfilzomib, and melphalan [22].

### 2.5. iPS/ES Genes

*SOX2* is a member of the *SOX* gene family and belongs to the B1 subgroup of the *SOX* family B group. *OCT4* is an octamer transcription factor, mainly expressed in totipotent embryonic stem cells and germ cells and downregulated during cell differentiation [64,65]. *SOX2* and *OCT4* can regulate the initiation of tumors and the function of CSCs [66,67]. *SOX2* and *OCT4* are highly expressed in SP cells [17], and *SOX2* and *OCT4* mRNA knockdown reduce the proportion of SP cells, suggesting that these factors are necessary to maintain SP content in MM cells. *SOX2* and *OCT4* may be a clinical drug target for MM. The expression of SOX2 in RPMI-8226 can be suppressed by cotylenin A [68]. In addition to *SOX2* and *OCT4*, iPS/ES genes such as Nanog may also be used as MM stem cell-like population markers, and further study is needed.

### 2.6. BTK

BTK, a key target in MM drug therapy, has a positive regulatory effect on stem genes (*OCT4*, *SOX2*, *NANOG*, and *MYC*) through the Akt/Wnt/β-Catenin pathway and enhances the self-renewal ability of MM cells. The overexpression of *BTK* in myeloma cells can increase their clonogenic ability and drug resistance, while inducible knockout of *BTK* abolished them [17]. In addition, the expression of stem genes such as *OCT4*, *SOX2*, and *NANOG* can be upregulated by BMSCs through the BTK pathway and increase MM clonogenicity [69].

### 2.7. RARα2

The expression of *RARα* increased significantly when MM recurred. The survival rate of MM patients with *RARα2* expression was significantly decreased, and RARα2 knockdown could significantly induce cell death and growth inhibition [18]. Besides, *RARα2* was also overexpressed in multiple myeloma stem-like cells. Overexpression of *RARα2* in MM cell lines can lead to increased drug resistance and clonal potential, activation of stem pathways (Wnt and Hedgehog pathway), increased SP cells ratio, ALDH, and stem gene expression [70].

### 2.8. ROS

It has been proved that cancer stem cells in hematology neoplasms, such as AML, have a low level of ROS, relatively [71,72,73]. In MM, evidence also shows that SP cells have a lower level of ROS than NSP cells, which means ROS may be a stem-cell marker in MM [29]. This may be a new direction for exploring MM stem cells.

In the study of MM stemness modulation, some molecules show no difference between MM cells and MM stem-cell-like cells, and then they may not be used in MM stem-cell-like population identification. However, they are still possible targets, including USP1, Wee1, and CD44, etc. USP1 siRNA knockdown can reduce the survival ability of multiple myeloma cells. The USP1 inhibitor SJB selectively blocked the activity of USP1. SJB also reduced the viability of multiple myeloma cell lines and patients’ tumor cells, inhibited the growth of multiple myeloma cells induced by bone marrow plasmacytoid dendritic cells, and overcame BTZ resistance [74]. High expression of Wee1 indicates poor survival in MM, and CD138^+^ plasma cells in MM patients have high sensitivity to MK1775 (Wee1 inhibitor) [75]. CD44 is a surface marker of breast, gastric, and colon cancer stem cells [76], and CD44 is closely related to the cell adhesion-mediated drug resistance of MM cells [77,78], which suggests that CD44 may also serve as a marker of MM stemness. Targeting these molecules is a potential treatment for inhibiting stemness and overcoming resistance in MM.

The characteristics of MM stem cell-like population and corresponding inhibitors were listed in Table 1.

## 3. Active Signaling Pathways Related to MMSCs and Drugs Targeting Them

Currently, the pathways involved in the stemness of MM include the Wnt/β-catenin, Hedgehog, Notch, and PI3K/Akt pathways, which play a vital role in a variety of cancer stem cells [79,80,81,82]. The following will describe the status of these signaling pathways in MM and the current drugs for these pathways.

### 3.1. Wnt/β-Catenin

Wnt signaling pathway is a group of multi-downstream channel signal transduction pathways, which plays a role in various stages of tumor formation. In the past, many Wnt pathway inhibitors have appeared and been used in various clinical tumor trials [83,84]. In MM, abnormality in the Wnt pathway is often caused by genetic or epigenetic mutations in the Wnt-regulating components, which plays an important role in the pathogenesis of MM and is likely to be an important therapeutic target for MM [85].

Drugs targeting the Wnt signal pathway have been developed frequently. Interestingly, Wnt3a can enhance the ratio of ALDH1+ cells by enhancing the expression of the β-catenin protein, but lycorine can inhibit ALDH1^+^ cells through Wnt/β-catenin pathway inhibition [42]. There are similar inhibitors, such as GSK126 [28], resveratrol [86] and BC2059 [87]. These studies indicate the significant role of the Wnt pathway in MM, and inhibition of the Wnt/β-catenin pathway is a potential method to inhibit the growth of MMSCs.

### 3.2. Hedgehog

Hedgehog (Hh) belongs to the intercellular signaling family, which plays an important role in the regulation of embryonic/somatic stem cells and the development regulation of tissues and organs. Alonso et al. proposed that Hedgehog and retinol-like signals could alter the microenvironment of MM by upregulating stromal CYP26, which helps to maintain a retinoic acid-low (RA-low) microenvironment, prevent differentiation and causing the resistance of BTZ [88]. Martello M et al. also proposed that controlling the Hh pathway in MM to prevent its overactivation might play a role in preventing MM recurrence. The researchers showed that patients could be divided into two groups according to the expression level of the Hh gene, among which patients with excessive activation of plasma cell Hh had poor prognosis and survival period, suggesting the promoting effect of Hh activation on MM recurrence [89]. Knocking down the GLI, signal transduction protein of hedgehog can cause the rate of SP cells to decrease [90]. These studies show that the Hedgehog pathway is critical in maintaining MM stemness and drug resistance but inhibitors need to be developed.

### 3.3. Notch

Notch is a transmembrane protein that can act as a receptor, as well as regulate cell transcription. The activation of Notch signaling is related to the pathogenesis of leukemia, lymphoma, MM, and other hematologic tumors [91,92]. The Notch signal pathway is important in MM chemotherapy resistance. Subsequent studies have shown that the activation of the Notch pathway plays a major role in myeloma cell resistance mediated by bone marrow stroma, and the use of γ-secretase inhibitor (GSI) can specifically inhibit the Notch pathway, reducing bone marrow stroma-mediated resistance and make myeloma cells sensitive to chemotherapy [93]. Jagged1-induced Notch activation has been shown to contribute to the BTZ resistance of myeloma cells in in vivo experiments [94]. Recent studies have shown that the inhibition of Jagged ligands can reduce bone marrow stromal cell-induced drug resistance (BTZ, lenalidomide, melphalan), suggesting that the Notch pathway is likely to be involved in MM resistance [95].

### 3.4. PI3K/Akt

The PI3K signaling pathway is extremely important for the growth of tumor cells [96,97], and its activation can lead to the inactivation of multiple tumor-suppressor genes. Multiple growth factors of myeloma cells can also play a role through the PI3K/Akt/mTOR pathway. Moreover, the activation of PI3K/Akt/mTOR is an important cause of bone lysis in MM patients [98]. Recent studies have found a positive correlation between the expression of ABCG2 on the surface of the side population of MM cells and the activation of PI3K/Akt/mTOR [99]. It has been demonstrated that microRNA-451 could regulate the PI3K/Akt/mTOR signaling pathway in multiple myeloma and then contribute to the stemness of side population cells [13]. The activation of this pathway has also been shown in bone marrow specimens of MM patients, which indicated that miR-205-5p could target RUNX1 and inhibit the activation of the PI3K/AKT/mTOR pathway to inhibit MM cell apoptosis [97,98]. In summary, the PI3K pathway is shown to be important for MMSCs.

Despite the related pathways of the MM stem-cell-like population not being fully clear, the above pathways have been proven to be closely related to MM stemness. The discovery and development of drugs targeting these pathways have potential clinical value for the treatment of MM. Figure 1 is a look at how these signaling pathways work in MM and the drugs that currently target them. And Table 2 shows the function and possible inhibitor of these pathways in MM.

## 4. Epigenetic Regulation of MM Stem Pathway

In 2015, Lydia HOPP et al. conducted a multiomics analysis of transcriptome and methylome data for B-cell lymphoma and pointed out that promoter methylation and histone modification can regulate the corresponding dry gene expression [109]. Epigenetic regulatory proteins (such as HDAC [110]) are also considered new targets for MM therapy. One objective is to explore the current therapeutic prospect of histone deacetylase inhibitor panobinostat in relapsed/refractory MM. In addition, the drug resistance and tumorigenicity of MM are regulated by non-coding RNA [111,112]. There are several non-coding RNAs that can regulate the activity of the previously mentioned stem-cell pathway. For example, miR-125b can regulate the expression of Notch1 and the activation of the Notch pathway in MM cells [105]. In a study by E Morelli et al., miR-125b-5p mimics can regulate the expression of IRF4, prolonging the survival time in a MM nude mice model. MicroRNA-451 regulates the stemness of side population cells via the PI3K/Akt/mTOR signaling pathway in multiple myeloma [13]. The high expression ofmiR-215-5p can inhibit MM cell apoptosis by targeting RUNX1 and inhibiting the activation of the PI3K/AKT/mTOR pathway [97,106].

## 5. Dispute on MM Stem Cells

Different from the definition of a stem cell, the definition of a cancer stem cell not only emphasizes its unlimited proliferation ability and differentiation into different types of cells but also its tumor-initiation effect and insensitivity to treatment [113,114]. In fact, the concept of MM tumor stem cells was proposed to explain the recurrence and reburning of MM cells, that is, although the tumor cells in MM patients are almost completely killed, the patients will still relapse [7]. At that time, the researchers called the small amount of MM cells with high tumorigenicity and drug resistance after treatment the MM stem cells. Although these cells have high tumorigenicity or stem pathway activation, there is no direct evidence that there are single tumor stem cells that can differentiate into various tumor cell groups in MM tumors. **However, research on these residual cells is of great significance to the development of clinical therapy. Follow-up researchers continue to explore and improve their marker characteristics and functional characteristics (Table 3).**

## 6. Conclusions and Future Perspectives

CSCs are a kind of cell that determines the occurrence, development, and metastasis of tumor tissue and has three characteristics: self-renewal, differentiation, and infinite proliferation. At present, CSCs have been found and defined in breast cancer, liver cancer, lung cancer, ovarian cancer, and other solid tumors. The surface markers include CD133, CD90, CD176, EpCAM, and so on [115,116]. The discovery of these surface markers makes the separation and screening technology of some solid CSCs capable, which provides important conditions for the research and development of anti-cancer drugs and on the elaboration of the mechanisms behind cancer resistance and recurrence.

There are similarities and differences between the markers of MM stem-cell-like cells and solid tumor stem cells. According to Table 1, MM cells with ALDH and ABC transporters are most likely to contain an MM stem-cell-like population, while CD24, CD44, and SOX2 play an auxiliary role in MM stem-cell-like population screening. ALDH and ABC transporter are markers of colorectal CSC, lung CSC and so on [117]. CD44 is used to screen more solid CSC than ALDH and ABC transporter, but its position in MM stem-cell-like population screening is not as good as theirs [118]. CD133 is also considered a marker for many solid tumor stem cells, but there is no evidence that CD133 can be used as a marker for a MM stem-cell-like population [118].

Although the definition of MMSCs (multiple myeloma stem cells) is controversial at present, the existing evidence shows that there are tumor-initiating cells in the MM cell population. These cells are closely related to the development, drug resistance, and recurrence of MM. Therefore, the definition, screening, and targeted inhibition of these cells are very important. In this paper, we list the markers used to screen the MM stem-cell-like population and the research into related inhibitors in MM treatment and introduce the pathways related to stemness in MM and the main target molecules in these pathways. At present, the mainstream view is that MMSCs are located in SP cells or the ALDH1^+^ cell population, which replaces MMSCs for research. Interestingly, BTZ can significantly inhibit the rate of SP cells but enhance the rate of ALDH1^+^ cells [42,119]. SP cells and ALDH1^+^ cells may represent different populations. However, we are not sure whether all MMSCs have these two biomarkers or whether they have other biomarkers. Although the MM stem-cell-like population is one of the causes of MM resistance, other factors are not excluded, such as cell-adhesion-mediated resistance, genetic-abnormality-mediated resistance, cell apoptosis, aging, DNA-repair-mechanism-defects-mediated resistance, and metabolic-changes-induced resistance. The mechanism of MM recurrence is also complex. It involves the tumor microenvironment and immune status, which is far from the explanation of MM stem cells.

Except for proteasome, the CD38 monoclonal antibody (daratumab) has also been applied to multiple myeloma, constantly enriching the therapy of MM patients. However, there are some drug resistance problems in the treatment of Dara, which may be related to various factors, such as the decrease of CD38^+^ immune cells [5,120]. If multiple types of monoclonal antibodies can be developed and used alternately in the clinic, it is likely to alleviate the clinical problem of CD38^+^ immune-cell reduction caused by the repeated use of CD38 monoclonal antibody alone. Several stemness markers (such as CD24 and ALDH1) mentioned in this paper are possible targets for the development of new monoclonal antibody drugs.

In recent years, the effect of epigenetic regulation on MM stemness has gradually become clear. HDAC and several miRs have been identified. In fact, the current research on Mir imbalance in MM has been quite in-depth, and the use of nanocarriers to deliver Mirs in the treatment of MM has a certain development value [121]. However, the Mirs mentioned in this paper are only verified at the cellular level. The function of these Mirs in MM animal models and primary MM patients has not been fully verified, so further research is needed.

## Figures and Tables

**Figure 1 cancers-13-03523-f001:**
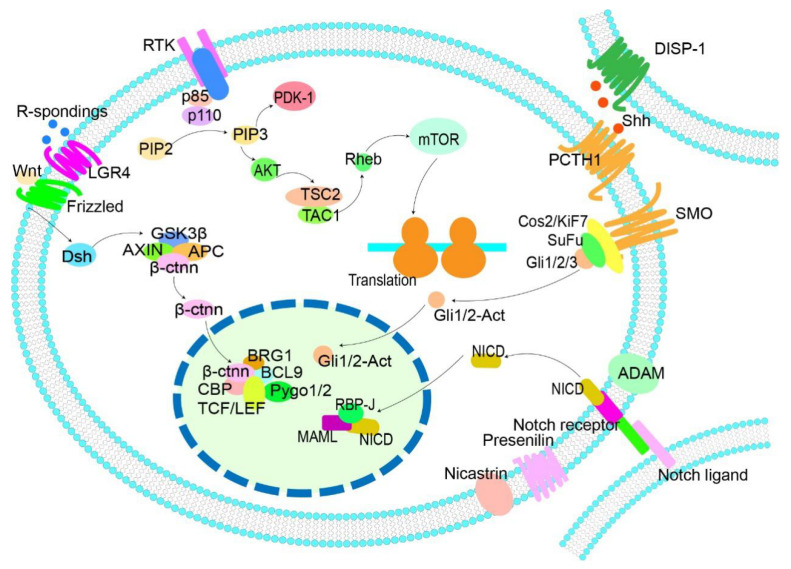
Known pathways related to stemness in MM. Dsh, disheveled protein; APC, adenomatous polyposis coli; β-ctnn, β-catenin; PI3K, phosphatidylinositol-3-kinase; DISP1, dispatched 1; SHH, sonic hedgehog; PTCH1, patched1; ADAM, a disintegrin and metalloprotease; NICD, Notch1 intracellular domain; RBP-Jk, recombination signal-binding protein-Jkappa; MAML, mastermind-like family members. This image describes the common pathways related to stemness and corresponding targets in MM cells. The Wnt, Notch, and hedgehog pathways can regulate MM cells at the transcription level, and PI3K at the translation level, and all of them are involved in the regulation of cell growth, survival, and drug resistance. In the Wnt pathway, GSK3 β protein expression was upregulated, and Wnt activation was inhibited on condition of PCDH10 gene overexpression. Drugs such as piceatannol, bc2059, imaquinone, and ethyl monoquinone can downregulate β-ctnn protein and inhibit Wnt pathway. R-spondings can act on lgr4 and promote Wnt pathway activation. In Hedgehog pathway, Shh and Gli1 can regulate bcl-2. In the Notch pathway, sahn1 can bind to the intracellular domain ICN of Notch and affect the recruitment of MMAL. GSI (an enzyme inhibitor) can act on γ-secretase and cause MM cell toxicity. In the PI3K/Akt/mTOR pathway, buparlisib can act on P110, while nvp-bez235 (imidazoloquinoline derivative) can bind to the ATP binding slit of PI3K and mTOR kinases and inhibit this pathway. These are the drug targets that have been studied.

**Table 1 cancers-13-03523-t001:** Markers of the MM stem cell-like population and inhibitors of them.

Surface Marker	Function	Drug	Mechanism	Reference Study
SP cells/ABC transporter	Trans-membrane transportation; related to drug resistance of CSCs; regulating oxidation reduction status; regulating membrane lipid composition; regulating the release of nutrients and metabolites, and regulating the tumor microenvironment	HLA class I small molecule antibody Lenalidomide Ibrutinib	Targeting β-catenin and suppressing stem genes such as *SOX2*, *OCT3/4*, and *Nanog*. Affecting phosphorylation of AKT, GSK-3-α/β, MEK1, c-JUN, P53, and P70S6K Targeting Bruton’s tyrosine kinase	[10,14,17]
		EZH1/2 double inhibitor (or-s1)	Targeting EZH1 and EZH2, causing Wnt signaling repression	[27]
		HuCD26mAb	Specifically inhibiting SP cells	[29]
		Fenretinide	Targeting IL-3, blocking pDC-induced MM cell proliferation	[30]
			Targeting CD26 in SP cells	[31]
ALDH	Promoting the dehydrogenation of acetaldehyde, participating in cell detoxification through oxidation, regulating the differentiation, apoptosis, and growth of CSC through the Ra signal pathway, and reducing the ROS in CSCs	GSK126	Abrogating the methylated histone 3 level, blocking the Wnt/β-catenin pathway, and inhibiting of EZH2 methyltransferase activity	[28]
		Lycorine	Inhibiting ALDH1^+^ cells through the Wnt/βcatenin pathway	[42]
		I-5-iodo-4′-thio-2′-deoxyuridine	Decreasing the activity of ALDH	[44]
		KS99	Targeting leukemia stem cells with high aldehyde dehydrogenase activity and inhibiting STAT3 phosphorylation and inhibiting the activation of Bruton’s tyrosine kinase.	[48]
		Disulfiram/Cu	Targeting ALDH1A1, inhibiting the expression of *NANOG* and *OCT*, and suppressing the Hh pathway by inhibiting transcription factors Gli1 and Gli2.	[49]
		MK1775	Inhibiting ALDH1^+^ cells through Wee1 kinase	[75]
CD24	Cell adhesion protein, mediating B cell antigen-dependent activation, distinguishing pre-B cells from Mature B cells, and overexpressed in SP cells	SWA11(CD24 antibody)	Targeting CD24	[22]
SOX2	Related to tumor invasion, metastasis, and EMT; an important index in clinical trials at present	cotylenin A and vincristine	Inhibiting *SOX2* mRNA expression in myeloma cells	[68]
CD44	Necessary medium for the bone-marrow adhesion of MM cells; participating in cell-adhesion-mediated drug resistance	All-trans retinoic acid (ATRA)	Downregulating the expression of total β-catenin, cell surface, and total CD44 in a mice xenotransplantation model; decreasing lenalidomide-resistant MM cells’ adhesion; and enhancing the effect of lenalidomide.	[78]

**Table 2 cancers-13-03523-t002:** Common pathway related to stemness in MM and corresponding drugs/regulators.

Pathway	Function	Drug/Target Gene	Mechanism	References
Wnt/β-catenin	Mediating the proliferation, migration, and drug resistance of MM cells; promoting the differentiation of osteoblasts	Resveratrol	Downregulating the expression of lncrna-neat1 in MM cells by suppressing the Wnt signaling pathway and UPR	[86]
		BC2059	Downregulating β-catenin protein	[87]
		Piceatannol	Decreasing the level of β-catenin, the transcriptional activity of Tcf4/lef complex, and the level of its target gene Axin 2	[100]
		Tumor suppressor gene PCDH10	Inhibiting the nuclear localization, lef/TCF activity, bcl-9, and Akt expression of β-catenin	[101]
		ilimaquinone and ethylsmenoquinone	Decreasing the level of β-catenin in the cell	[102]
		LiCl	Inducing G2/M phase arrest of the MM cell cycle; activating the Wnt/β-catenin signaling pathway to induce MM cell apoptosis	[103]
Hedgehog	Changing the tumor microenvironment, participating in BORTEZOMIB resistance, inhibiting the apoptosis of MM cells	CYP26	Forming a low retinoic acid environment and producing resistance to BTZ	[88]
		SHH (sonic hedgehog)	Inhibiting the apoptosis of myeloma cells	[104]
Notch	Participating in the development of MM, related to strom-mediated drug resistance.	GSI (γ-secretase inhibitor)	Inhibiting the second cleavage of Notch receptor	[93]
		miR-125b	Targeting MALAT1 and regulating the proliferation of MM cells	[105]
PI3K/Akt	Tumor-suppressor gene inactivation, related to bone lysis	miR-215-5p	Targeting RUNX1 and inhibiting the PI3K/AKT/mTOR pathway	[106]
		miR-30d	Targeting metaherherin and inhibiting the PI3K/Akt signal pathway	[107]
		NVP-BEZ235	Binding to the ATP binding gap of PI3K and mTOR kinases	[108]

**Table 3 cancers-13-03523-t003:** Biological function characteristics of MM with specific markers.

MM with Specific Markers	Biological Function Characteristics	References
Side population cell	High clonogenicity, tumorigenicity, and self-renewal ability	[14,15]
ALDH1^+^ MM	High colony-forming ability, resistance to bortezomib and adriamycin	[40,41]
CD138^−^ MM	Strong colony-forming and tumor initiation ability	[58]
CD24^+^ MM	Strong colony-forming ability and tumorigenicity, high iPS/ES genes expression, and strong resistance to MM clinical drugs	[22]

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
