# Peer review of "Identification and Characterization of Multiple Myeloma Stem Cell-Like Cells"

_cancers, 2021, doi:10.3390/cancers13143523_

Round 1

Reviewer 1 Report

Identification and Characterization of Multiple Myeloma Stem 2 Cell-like Cells

According to the authors, the cells described in this review are closely related to the drug resistance and relapse of multiple myeloma (MM) and have characteristics of cancer stem cells. The authors’ goal is to introduce the latest developments in ‘MM stem cell-like cells’, list possible markers for these cells, introduce the mechanism by which ‘MM stem cell-like cells’ affect MM resistance and recurrence, and explain the role of biochemical pathways that are related to ‘MM stem cell-like cells’.

Comments and Concerns

This review is essentially an encyclopedia of information about a multitude of different genes expressed in MM tumors and the effects of various drugs on their expression.  That information will be useful to researchers working on MM cancers. 

The title of this review is misleading; it should read “Identification and Characterization of ‘Side Population Cells’ from Multiple Myelomas”.  Side population cells’ (section 2.1) were identified in 1996 and - based on this review - the evidence that SP cells are MM stem cells is purely circumstantial.  In fact, the evidence presented here supports the conclusion that the primary function of SP cells is to exporting drugs from the tumor, thereby promoting resistance to chemotherapy.

The Introduction states, “Although the existence and identification of MM stem cells are controversial at present, the existing data show that there is a kind of progenitor cells with low differentiation in the MM cell group which has the characteristics of CSC and so we call them ‘MM stem cell-like cells’.”  Explain the controversy. 

What is the evidence for and against the hypothesis that MM tumors contain ‘cancer stem cells’?  Can SP cells differentiate into other types of cells commonly found in MM tumors when they are cultured in vitro?  Do SP cells produce tumors when inoculated into immune-compromised mice?  If so, do they give rise to all of the other types of cells found in MM tumors?  Do any non-SP cell lines give rise to tumors in immune-compromised mice?  A recent review of MM cell tumorigenesis does not appear to mention SP cells (“Evaluating the efficacy of multiple myeloma cell lines as models for patient tumors via transcriptomic correlation analysis”, Vishesh Sarin et al., Leukemia 34, 2754–2765 (2020) PMID: 32123307).

The fact that cancer cells from a variety of different types of cancer frequently express one or more genes that are associated embryonic stem cells or induced pluripotent stem cells does not make these cancer cells stem cells.  This review would benefit significantly if the authors delved into the controversial aspects of cancer stem cells in MM tumors.

The reason that the existence of multiple myeloma cancer stem cells is controversial, is that researchers do not distinguish between and stem cells, progenitor cells, and simple cell variations that exist within every cell culture.  The term ‘cancer stem cell’ is currently fashionable, but often in appropriate.  The term cancer stem cell-like cell only adds to the confusion. 

Stem Cells are cells that can reproduce indefinitely without losing their ability to differentiate in response to environmental signals into all of the different cells that make up a specific tissue, organ, or organism.  The paradigm for stem cells in mammals is “embryonic stem cells”.  They can be cultured in vitro without losing their ability to differentiate into cells derived from all three primordial germ layers: ectoderm, mesoderm and endoderm.

Progenitor Cells are simply cells from which another cell type is descended.  Cell differentiation was recognized long before the discovery of ‘stem cell populations’ that were maintained within the tissue, organ, or organism.

Culturing cells in vitro will select for cells that “have stronger proliferation, migration, tumorigenesis and drug resistance than general tumor cells”.  Such cells are simply ‘robust MM cells’.  Cells derived from cancers generally exhibit one or more stem cell characteristics, produce tumors (benign or malignant) when introduced into immune-compromised mice, and differentiate due to their genetic instability.  However, they are neither ‘stem cells’ nor ‘stem cell-like cells’; they are ‘progenitor cells’.

The manuscript would benefit by proofreading grammar and syntax.

Author Response

Evidences for and agains the existence of multiple myeloma cancer stem cells

Thank you for reviewing this paper in your busy schedule. We noticed that the Introduction states, “Although the existence and identification of MM stem cells are controversial at present, the existing data show that there is a kind of progenitor cells with low differentiation in the MM cell group which has the characteristics of CSC and so we call them ‘MM stem cell-like cells’.”. Thank you for reminding us to supplement evidences for and against the hypothesis that MM tumors contain “cancer stem cells”. The seventh reference of this paper (Huff, C.A.; Matsui, W. Multiple myeloma cancer stem cells. J. Clin. Oncol.2008,26, 2895-2900.), a review of MMSC stated the evidences. It is stated that clonotypic B cells have the ability to reproduce multiple myeloma in immunodeficient mice, and these cells can self-renew, differentiate into effector cells (plasma cells) and have drug resistance. In these characteristics, clonotypic B cells are similar to stem cells. The evidence against the presence of cancer stem cell was also mentioned in Huff’s review, “the exact phenotype of the clonogenic cell has not been definitively established and controversy remains.” Moreover,   a review(Abe M, Harada T, Matsumoto T. Concise review: Defining and targeting myeloma stem cell-like cells. stem Cells.  2014,6(38): 40496- 40506. )and a article  (Jin N, Zhu XJ, Cheng FJ.Disulfiram/copper targets stem cell-like ALDH+ population of multiple myeloma by inhibition of ALDH1A1 and Hedgehog pathway. J Cell Biochem. 2018,119(8): 6882-6893.) of MMSC stated the evidence.

 About SP cells

SP cells can differentiate into non-SP cells, and can also become tumors in immune deficiency. However, non SP can also differentiate into SP and form tumor to a certain extent, but it is much weaker than SP cells.

As for the recent paper (“Evaluating the efficacy of multiple myeloma cell lines as models for patient tumors via transcriptomic correlation analysis”, (Vishesh Sarin et al. Evaluating the efficacy of multiple myeloma cell lines as models for patient tumors via transcriptomic correlation analysis. Leukemia, 2020,2754-2765.), this article does not mention SP cells, because SP cells are only a special population (with ABC transporter expression) in MM cell lines. Vishesh et al's research is mainly to compare and evaluate the advantages and disadvantages of several MM cell lines as experimental models at the transcriptome level, mainly to compare MM cell lines with primary cells of MM patients, Therefore, SP cells in MM cell lines were not screened.

We agree with your views on stem cells and tumor initiating cells, and thank you for your guidance on the term "multiple myeloma stem cell like cells" in this article. The cells with specific surface makers described in this paper (such as CD24+ MM cells), are similar to the side population cells in the phenotype of drug resistance, but are essentially different from the side population cells. The definition of side population cells is very clear, that is, they have the function of dye efflux, and they are distributed in the cell population next to the main population in the process of flow cytometry. However, the markers such as CD24 in this paper do not have the function of dye efflux, and the mechanisms involved in the tumorigenesis and drug resistance of these cells are complex, which can't be explained by the function of these markers. In addition, the side population cells may also have other markers mentioned in this paper. Therefore, "side population cell" is not suitable for summarizing the cells with other marker characteristics mentioned in this paper.

Different from the definition of stem cell, the definition of cancer stem cell not only emphasizes its unlimited proliferation ability and differentiation into different types of cells, but also emphasizes its tumor initiation effect and insensitivity to treatment. “Historically, it first invoked the concept that malignant cell populations are organized as unidirectional cellular hierarchies in which CSCs constitute biologically unique subsets of cells, which are distinguished from the bulk of the cells that they produce by their exclusive ability to perpetuate the growth of a malignant cell population indefinitely”, Peter et al mentioned in Nature Reviews cancer. They also suggested that CSC should only be used in malignant and aggressive tumors, and the significance of CSC therapy is emphasized. “The tumour may initially appear to be eliminated, but it later reappears because the rarer and grossly invisible populations of CSCs have survived.”To a certain extent, the definition of cancer stem cell is proposed to explain the clinical phenomenon of recurrence in many patients after treatment (Valent, P., et al. Cancer stem cell definitions and terminology: the devil is in the details. Nat Rev Cancer, 2012,12(11): 767-75.) CSC, properly known as cancer transmitting cells, is defined by two attributes: self-renewal and pluripotency. At present, transplantation experiment and continuous tumor transplantation experiment are the gold standard of CSC identification(Rycaj, K. and D.G. Tang. Cell-of-Origin of Cancer versus Cancer Stem Cells: Assays and Interpretations. Cancer Res, 2015, 75(19): 4003-4011.)

As you suggested, the cells we described are just robust tumor cells. As far as the current definition and identification criteria of CSC are concerned, CSC, in a sense, is also a robust tumor cell with strong tumorigenicity. For example, if we inject a tumor cell into nude mice and successfully build a transplanted tumor model containing multiple cell subsets, then there is no doubt that the tumor cell is a tumor stem cell. Compared with ordinary cancer cells, a smaller amount of cancer cells (with the characteristics in our review)can developed into multiple myeloma in nude mice. Naturally, tumor initiating cells tend to exist in this kind of population. However, it cannot be ruled out that there are no tumor initiating cells in the cells without these characteristics. Therefore, we call the cells with the characteristics of the markers mentioned in this paper "multiple myeloma stem cell like cells" because of the lower therapeutic sensitivity and the activation of some stem cell pathways and because they may have a higher proportion of cancer stem cells. The significance of this concept is that in the study of MM disease initiating cells, some scholars regard the cells with the characteristics of these markers as tumor stem cells. In fact, these cells are not tumor stem cells, but tumor stem cells tend to exist in the cell population with these markers.

In order to avoid the misleading, we add a dispute on definition of multiple myeloma cancer stem cell, which was proposed by huff et al.(Huff, C.A; Matsui, W. Multiple myeloma cancer stem cells. J. Clin. Oncol. 2008,26,2895-2900.). Some evidences supporting and not supporting the existence of MMSCs are listed.

The language problem

Thank you for your suggestion. We have carefully revised the manuscript according to the reviewers' comments.

Reviewer 2 Report

Wancheng Guo et al reviewed the latest developments in multiple myeloma - MM stem cell-like cells (MMSCs), uncover the mechanism of how MMSCs work in MM resistance and recurrence, and explain the role that active pathways related to stemness of MM. Point to be considered:

1) The rationale of why the authors came up with this review.

2) What is the information that is not exactly available that motivated the authors to come up with this information. What are the current caveats and how do the authors highlight the current research in answering them? If not they need to address in future directions..

3) The authors need to deeper describe the role of epigenetics: they themselves mentioned (reff-105-107) some seminal manuscript in this regard. Nonetheless, this reviewer personally misses some important pieces of evidence regarding miRs delivery systems including liposomes, polymers, and exosomes that increase their physical stability and prevent nuclease degradation. Phase I/II clinical trials support the importance of miRs as an innovative therapeutic approach in nanomedicine to prevent cancer progression and drug resistance (mainly putatively mediated also by MMSCs, refer to PMID: 32349317). Please expand in discussion.

4) The authors need to highlight what new information the review is providing to enhance the research in progress

5) Can the author provide additional eminent example that have been already attempted and might foster future studies, a challenge, but with the potential to increase the interest for a broad readership in the field (i.e., novel MoAb -an also BiTEs and CAR-T -clompletely changed MM scenario. As CD38 is expressed on several immune cells, Daratumumab (dara) treatment depletes CD38+ immune cells, causing a modification of the antitumor response. Dara treatment reduces the immunosuppressive cells in MM microenvironment, including myeloid-derived suppressor cells (MDSCs), Treg, and Breg cells, and increases the anti-MM activity. Dara treatment depleted CD19+C24+CD38+ Bregs in MM patients immediately after the first infusion and during the entire regimen, as well as when the Dara-treated patients relapsed (PMID: 31936617). Do the authors would be able to integrate their perspective of how more individualized approaches need to be tested in well-designed clinical trials also taking into consideration MMSCs?

Author Response

1) The rationale of why the authors came up with this review.

In 2008, huff et al. published a review of "multiple myeloma cancer stem cell". So far, there is no specific molecular phenotype of stem cells in multiple myeloma. Therefore, the term "cancer stem cell" is not rigorous. It can only be said that MM stem cells are more likely to exist in the "cancer stem cell" mentioned by huff et al, It is called "cancer stem cell like cell" in this paper. The basic research of this kind of cells is largely based on the clinical background of relapse and re-ignite of MM patients.

Follow up studies show that cells with a variety of molecular phenotypes have the characteristics of "cancer stem cell like cell". This paper lists the identification indicators of surface markers proposed in recent years, and summarizes the activation of stem cell pathway in MM cells. For these surface markers and pathways are also important therapeutic targets for MM, we summarize the targeted inhibitors for these molecules in recent years.

2)  What is the information that is not exactly available that motivated the authors to come up with this information. What are the current caveats and how do the authors highlight the current research in answering them? If not they need to address in future directions.

The current controversy is that there is no unified standard for the screening of MM stem cell like cells, and the markers we listed can be used as a reference for screening and identification. This paper introduces many specific MM cell populations with stemness characteristics, which have high tumorigenicity, drug resistance and activation of some stemness pathways. Therefore, inhibitors targeting these cells are of great significance for the treatment of MM patients, and it may be one of the methods to use the stem cell pathway as a target for drug development. At the same time, we add the controversial point about the term "MMSC" in the article.

3) the role of epigenetics

Thank you for your suggestion. We added a paragraph about epigenetic regulation of stemness after the stemness pathway, introducing the role of HDAC and several miRs to stemness. The “discussion” part was transformed into “Conclusions and Future Perspectives”, we propose some discussions about miR in MM stemness regulation.

4)  The authors need to highlight what new information the review is providing to enhance the research in progress.

As is mentioned in the “Conclusions and Future Perspectives” part, the main aspect of the review is focusing on the controversies surrounding the identity of cancer stem cells in the disease. In addition, we think that sorting MMSC is a good method to evaluate the therapeutic effect of drugs on MM, but the specific characteristics of MMSC should be further studied. The inhibitors targeting stem cell pathway in MM cells need to be further developed.

 5)  About the Dara therapy

As you mentioned, the Dara therapy, which targets CD38+ cells, is efficiency in some patients. So, we tentatively added the discussion about Dara resistance in the last part, and added the discussion about the possibility of combined therapy of multi-monoclonal antibodies targeting to MMSC surface markers, which may overcome the Dara therapy.

Round 2

Reviewer 2 Report

The authors have clarified several of the questions I raised in my previous review. Most of the major problems have been addressed by this revision.